# Different Combinations of Butchery and Vegetable Wastes on Growth Performance, Chemical-Nutritional Characteristics and Oxidative Status of Black Soldier Fly Growing Larvae

**DOI:** 10.3390/ani11123515

**Published:** 2021-12-09

**Authors:** Nicola Francesco Addeo, Simone Vozzo, Giulia Secci, Vincenzo Mastellone, Giovanni Piccolo, Pietro Lombardi, Giuliana Parisi, Khalid A. Asiry, Youssef A. Attia, Fulvia Bovera

**Affiliations:** 1Department of Veterinary Medicine and Animal Production, University of Napoli Federico II, 80137 Napoli, Italy; nicolafrancesco.addeo@unina.it (N.F.A.); vozzosimone@gmail.com (S.V.); vincenzo.mastellone@unina.it (V.M.); giovanni.piccolo@unina.it (G.P.); pilombar@unina.it (P.L.); 2Department of Agriculture, Food, Environment and Forestry, University of Florence, 50144 Firenze, Italy; giulia.secci@unifi.it (G.S.); giuliana.parisi@unifi.it (G.P.); 3Agriculture Department, Faculty of Environmental Sciences, King Abdulaziz University, P.O. Box 80208, Jeddah 21589, Saudi Arabia; kasiry@kau.edu.sa

**Keywords:** *Hermetia illucens*, growth, performance, vegetable mix, butchery wastes, hemolymph, larvae chemical traits

## Abstract

**Simple Summary:**

Due to the high sustainability of insect farming, the possibility to farm insects as a food and feed source seems to be very promising. Reusing and enhancing food waste is possible by using it as a substrate for the growth of insects. In this context, black soldier flies (BSF) can grow on a wide range of substrates, transforming them into valuable biomass. In this trial, four different substrates were used and were evaluated for their suitability for larvae rearing: broiler feed as standard diet, a vegetable diet, a diet with 50% of vegetables and 50% of butchery wastes, and a diet composed by 75% of vegetables and 25% of butchery wastes. Butchery wastes can be suitable, but they must be well combined with other ingredients to balance the high level of lipid and the low content of protein, and vegetable wastes can be an appropriate candidate. Vegetable and butchery wastes are easy to find and collect, and in the present trial, they showed interesting potential for BSF larvae growth producing, at 22 days of age, insects with interesting chemical characteristics. The use of vegetable wastes reduced the level of the reactive oxygen species in insect hemolymph, suggesting a positive effect of larvae welfare.

**Abstract:**

*Hermetia illucens* larvae (five days old) were farmed on broiler feed (control diet), a vegetable diet (V100), a 50% of vegetable diet + 50% of butchery wastes (V50 + B50), and a 75% of vegetable diet + 25% of butchery wastes (V75 + B25) to evaluate their suitability. Ten kilograms of substrate and 6000 larvae composed each replicate (nine per group). Larvae were weighed and measured every two days until the 25% developed into prepupae. Larval mortality and growing indexes were calculated. Substrates, larvae, and frass chemical composition were analyzed. Larvae oxidative status and stability were measured in hemolymph and body. The V100 larvae showed the lowest live weight, length, thickness, and growth rate but had low mortality rate and high substrate reduction index and protein conversion ratio. The V100 larvae had similar protein to and lower lipids than the control ones, while the V50 + B50 and V75 + B25 larvae contained higher lipids and lower protein than the others. Despite the vegetable wastes, at different levels, the reactive oxygen species content decreased in hemolymph, and the V100 diet depressed growth performance and should be avoided. The use of butchery wastes combined with vegetable ingredients can be a suitable alternative to balance the high level of lipid and the low content of protein.

## 1. Introduction

Global average temperature has increased by about 0.7 °C in the last century [1]. The Intergovernmental Panel on Climate Change (IPCC) reported that anthropogenic greenhouse gases (GHG), including carbon dioxide (CO_2_), methane (CH_4_), nitrous oxide (N_2_O) and halocarbons, have been responsible for most of the observed temperature increase. The effects of global warming are evident and have led to an increasing attention of the world population towards the environment and biodiversity; thus, a greater sustainability of anthropogenic activities is required.

Among the anthropogenic activities contributing to global warming, the livestock sector is under “special surveillance”. Firstly, Gerber et al. [2] estimated that the livestock sector contributes 14.5% of global GHG emissions, which are mainly responsible for climate change [3]. Secondly, the societal concern over animal welfare has increased according to the number of citizens preferring for farm animals to be treated as humanely as possible [4]. For these reasons, in recent years, several authors have studied the possibility of reducing the global impact of animal production.

Among the different solutions, the possibility of farming insects as a food and feed source seems to be very promising due to the high sustainability of insect breeding. In fact, insect farming requires less land and water and produces lower GHG emissions in comparison to traditional livestock productions [5]. In addition, insects have high feed conversion efficiencies and can transform low-value organic by-products into high-quality food or feed [5]. The industrial farming of insects, aiming to maximize mass production, very often uses industrial products as growing substrate for larvae, such as poultry diet. Widening the possibilities of using alternative and more sustainable substrates will play a key role in enhancing the circularity of insect production, helping European insect farms to reach their full potential [6].

In recent years, some authors starting to investigate potential new substrates for insects, mainly for black soldier fly larvae (*Hermetia illucens*) [7,8,9,10] obtaining interesting results in terms of growth performance and larval chemical quality. Black soldier fly is one of the most reared insects in the world, not only for its bioconversion ability [11] but also for the biological active molecules that can derive from it such as chitin [12], lipids [13], or antimicrobial peptides [14]. Due to the great potential of the black soldier fly, a wide range of substrates merits to be explored for its larval growth, in particular, wastes.

Butchery waste falls into category 3 of animal by-products, defined by Regulation (EC) 1069/2009 [15]. They include mainly fat, bones, and small amounts of meat, which, for only commercial reasons, cannot be used for human consumption. They must therefore be disposed of, and this operation has a cost. It could be interesting, with a view to a circular economy, to try to make the most out of this waste. The richness in fat of butchery waste could be “mitigated” using vegetable wastes also readily available and notoriously richer in water and carbohydrates.

Since no indications are available concerning the welfare of larvae and the potential effect of stress on the quality of the final product, a first step could be to explore the metabolic conditions of larvae growing on different substrates. The evaluation of the used substrates on oxidation index responses could provide complementary physiological information for the assessment of health and well-being outcome of larvae. It is difficult to quantify the reactive oxygen species (ROS) in practice due to their very short half-life, and it requires complex techniques over a long period of time [16]. Due to their high reactivity, ROS react with practically every organic molecule they meet, producing reactive oxygen metabolites (ROMs), which are more stable than the ROS and are therefore easier to quantify.

Conversely, the biological antioxidant potential (BAP) matches the total antioxidant capability of plasma and includes either exogenous (ascorbate, tocopherols, carotenoids) or endogenous (proteins, glutathione peroxidase, superoxide dismutase, catalase) components that can oppose the oxidant action of reactive species [17]. For these reasons, for the laboratory assessment of the oxidative status, we used the pro-oxidizing component, through the d-ROMs for the determination of plasma hydroperoxides, and the antioxidant component, through the BAP test for the evaluation of the total plasma antioxidant barrier [18].

The aim of the present research was to use wastes obtained from vegetable markets and from butcheries in different combinations as substrate for black soldier fly larvae to produce high quality larvae for feed. Three different waste combinations were compared to a standard diet with the purpose of testing what combination would give the best results in terms of larvae growth performance and chemical traits. In addition, the evaluation of the oxidative stress markers in hemolymph and the oxidative stability of larvae could supply interesting information about animal health when farmed on different utilized substrates for their growth.

## 2. Materials and Methods

### 2.1. Larvae and Substrates

Five-day-old black soldier fly larvae (*Hermetia illucens*) were purchased from the commercial insect rearing company Smart Bug’s (Treviso, Italy) in February 2020 and used in a growing trial. Four different substrates were used in the trial: broiler feed as a control diet; a total vegetable mix diet (V100); a diet consisting of 75% of vegetable diet + 25% of butchery wastes (V75 + B25); and a diet consisting of 50% of vegetable diet + 50% of butchery wastes (V50 + B50). The vegetable diet consisted of a mix of vegetable wastes collected from fruit and vegetable shops in the province of Napoli (Italy), containing 75% of vegetables (broccoli 40%, celery 35%, cabbages 25%) and 25% fruits (50% oranges and 50% apples). The butchery wastes were obtained from butchers in the province of Napoli and mainly consisted of fat and meat resulting from the trimming of bovine carcasses and cuts of meat. The collected vegetable wastes were stored for two days at room temperature to reduce the water content. Vegetables and butchery wastes were cut into small pieces prior to use. Diets were prepared mixing the ingredients accurately. A batch feeding strategy was applied, which means that the substrates were placed in plastic containers (60 cm × 40 cm × 15 cm) one day prior to placing the larvae, at the beginning of the experiment. This allows the substrates to heat up until the start of the experiment.

Each group (Control, V100, V50 + B50, and V75 + B25) consisted of 9 replicates, each placed in a plastic container for a total of 36 trays. In each replicate, 10 kg of substrate were placed. On the top of each substrate, 6000 five-day-old larvae were transferred after weighing. To calculate the average weight 100 larvae/time were counted and weighed on an analytical balance (Adventure Pro balance, Ohaus, Pine Brook NJ, USA) for a total of 60 weighings. The trays were covered with a perforated cap with a black nylon grid and placed in a ventilated chamber (air flow around 2 m/s) under controlled environmental conditions (T: 27 ± 0.5 °C; RH: 70 ± 5%; L:D photoperiod: 16:8).

Moisture content of the substrates was measured at the beginning of the trial on 10 g of each substrate, using an electric oven for 24 h at 65 °C. The water contents of the vegetable and butchery wastes were 85.37 ±  1.12% and 38.92  ±  2.85%, respectively. Thus, the water percentage in the four diets was 70.05, 85.37, 70.13, and 63.65%, for Control, V100, V75 + B25, and V50 + B50 diets, respectively. The plastic containers were visually inspected daily to verify the adequate level of humidity. In addition, the temperature in each container was recorded every day to verify the optimal conditions for the larvae.

### 2.2. Growing Trial

One hundred larvae per replicate were randomly selected every two days, weighed, and measured for length and thickness (measured at the equator of each larva) and were then returned to their respective container.

Feeding of larvae was continued until more than 25% of the larvae in a tray had developed into prepupae. The evaluation of the prepupae percentage has been undertaken by collecting exactly 100 g of substrate + larvae from each container (in three replicates) and counting the number of larvae and prepupae contained in each replicate.
Larval mortality (LM), % = (ILN − (FLN + FPN)) ⁎ 100/ILN
Growth rate (GR), % = (LFW, g − LIW, g)/d
Substrate reduction (SR), % = (AS, g − RS, g) ⁎ 100/AS, g
Waste reduction index (WRI), % = (AS, g − RS, g) ⁎ 100/AS, g/d
Efficiency of conversion of digested feed (ECD) = TFB, g/(TS, g − RS, g),
where ILN = initial larval number; FLN = final larval number; FPN = final prepupae number; LFW = larval final weight; LIW = larval initial weight; d = days of the trial; AS = administered substrate; RS = residual substrate; TFB = total final biomass; TS = total substrate; and RS = residual substrate.

All the weights are expressed on a dry matter basis.

Larvae yield (LY) was calculated as the ratio between larvae total biomass produced at the end of the trial and the total available substrate on a dry matter basis.

In addition, the protein conversion ratio (PR) was calculated considering the indications of Ewald et al. [19] as follows:

PR = total protein in final larval biomass/total protein in the substrate.

### 2.3. Chemical-Nutritional Characteristics and Oxidative Stability of Larvae

At the end of the trial, samples of substrate, larvae and frass from each tray were collected, freeze-dried using a Micromudulyo freeze drier (Thermo Electron Corporation, Thermo Fisher Scientific Inc., Whaltham, MA, USA) and analyzed for chemical composition. Dry matter (DM), ashes, and crude protein (CP) were analyzed according to AOAC [20]. In brief, for DM and ashes, around 2.5 g of sample were weighed into porcelain capsule and put in an electric oven at 103 °C until constant weight; then, the capsule was transferred to an electric stove at 550 °C for the whole night. The crude protein was determined using the Kjeldahl method; only for larvae, the nitrogen to crude protein conversion ratio was 4.76 according to Jansen et al. [21]. Total lipids were extracted from each sample according to the Folch et al. [22] method and gravimetrically quantified. The amount of carbohydrates (CHO) in the diets was calculated as follows: CHO, % DM = 100—Ash, % DM—CP, % DM—Lipids, % DM.

Oxidative stability of larvae lipids was analyzed following the primary and secondary oxidation products by means of conjugated dienes (CD) and 2-thiobarbithuric acid reactive substances (TBARS), determined according to the spectrophotometric methods previously proposed by Srinivasan et al. [23] and Vyncke [24], respectively. The analyses were performed in duplicate, and the results were expressed as mmol hydroperoxides (mmol Hp)/100 g larvae and malondialdehyde equivalents (MDA-eq.)/100 g larvae, respectively.

### 2.4. Oxidative Status of Larvae

At 20 days of age (15th day of the trial), hemolymph samples were collected from twenty larvae per replicate, according to Łoś et al. [25]. Briefly, the larvae were immobilized with tweezers, an incision of body layers was made with a scalpel, and the floating hemolymph was collected with a pipette and then frozen in a tube containing 150 µL of 0.6% physiological saline until analysis. d-ROMs and BAP tests were measured using reagents from Diacron International s.r.l. (Grosseto, Italy). In the d-ROMs test, reactive oxygen metabolites (primarily hydroperoxides) in a biological sample, in the presence of iron released from plasma proteins by an acidic buffer, are able to generate alkoxyl and peroxyl radicals, according to the Fenton reaction. Such radicals can then oxidize an alkyl substituted aromatic amine (*N,N*-dietylparaphenylendiamine), thus producing a pink-colored derivative which is photometrically quantified at 505 nm [26]. The d-ROMs concentration is directly proportional to the color intensity and expressed as Carratelli Units (1 CARR U = 0.08 mg hydrogen peroxide/dL). In the BAP test, the addition of a sample to a colored solution, obtained by mixing ferric chloride solution with a thiocyanate derivative solution, causes a discoloration, whose intensity was measured photometrically at 505 nm and was proportioned to the ability of the plasma to reduce ferric ions [17]. The results were expressed as μmol/L of reduced ferric ions.

### 2.5. Statistical Analysis

Data were analyzed by a one-way ANOVA, using the GLM procedure of SAS [27] and considering the substrate as main effect.

The experimental unit was the replicate. To assess the differences among means, Tukey’s test was used [27].

## 3. Results

All the groups reached the end of the experiment (25% of prepupae) after 17 days, when larvae were 22 days old. Considering the amount of administered substrate (10 kg), the number of larvae for replicate (6000), and the length of the growing period (17 days), the feeding rate in the present trial was around 0.098 mg of substrate per larva/d.

Table 1 shows the chemical composition of the substrates used in the trial.

The average live weight of BSF, measured on 100 larvae per replicate every 2 days throughout the trial, is reported in Table 2. In general, larvae of 75V + 25B group showed a higher live weight (LW) compared to the other groups, even if, at the end of the trial, they had a similar weight to that of the control group. At 22 days of age, larvae from the V100 group showed the lowest LW (*p* < 0.001).

Starting from 11 days of age, larvae from V75 + B25 and V50 + B50 groups showed a similar length and, at the end of the trial, were longer than the larvae of the control group, while larvae from the V100 group showed intermediate values (Table 3).

The height of larvae, measured in the middle of the body, is pictured in Table 4. In general, the V75 + B25 group showed the highest values up to 17 days of age. Then, the control group larvae overcame all the others. Starting from 11 days old, V100 groups showed the lowest height values.

The growth performance of the BSF larvae during the trial are summarized in Table 5. The V50 + B50 group showed the highest mortality rate (*p* < 0.01), followed by the V75 + B25 group and, together, control and V100 groups. The total larval biomass in the V100 group was the lowest (*p* < 0.01), while the V75 + B25 group had a higher total larval biomass than the control and V100 groups. The total larval frass showed the highest value in the control, followed by V50 + B50, V75 + B25, and V100 groups (*p* < 0.01). The length to height ratio of V50 + B50 group larvae was lower (*p* < 0.01) than that of V100 larvae and higher (*p* < 0.01) compared to the control group larvae. The growth rate of V100 group was lower (*p* < 0.01) than of the other groups; the opposite happened for the substrate reduction (SR) index. In addition, the control group showed the lowest (*p* < 0.01) SR value. Larvae yield obtained in the V50 + B50 group was lower (*p* < 0.01) than that of the V100 and V75 + B25 groups. The waste reduction index of V50 + B50 group was lower (*p* < 0.01) than that of V75 + B25 and higher than that of the control group. The efficiency conversion of digested food was the highest (*p* < 0.01) in the control group, followed by both V100 and V75 + B25 and then by the V50 + B50 groups. The protein conversion ratio was the highest (*p* < 0.01) in the V100 and V75 + B25 groups.

The hydroperoxide levels and the antioxidant capacities found in hemolymph of larvae fed different diets are indicated in Table 6. The antioxidant barrier was not significantly different among groups, whereas the concentration of hydroperoxides was higher in the control group, thereby showing that the use, as well as the inclusion, of vegetables in the diet accumulated less oxidative damages.

At the end of the trial, the larvae of V100 group showed the highest (*p* < 0.01) moisture, followed by the control and, together, the V50 + B50 and V75 + B25 groups. The V100 group showed the highest (*p* < 0.01) amount of ash, followed by the control, V75 + B25, and V50 + B50 groups (Table 7). The highest percentage of lipids (*p* < 0.01) was found in the V50 + B50 group and the lowest in the V100 one. The V50 + B50 group showed a lower percentage of protein (*p* < 0.01) compared to the control and V100 groups. The CD level in the control and V50 + B50 groups was higher (*p* < 0.01) than the other groups. The MDA of V75 + B25 group was higher (*p* < 0.01) than the values found in the other groups.

Regarding the chemical traits of frass (Table 8), the moisture was the highest (*p* < 0.01) in the V50 + B50 group, followed by V100, control, and V75 + B25 groups. The ash percentage was the highest (*p* < 0.01) in the V100 group, followed by the control, V75 + B25, and V50 + B50 groups. The highest lipid percentage (*p* < 0.01) was measured in the V50 + B50 group, while the lowest was in the control and V100 groups. The V100 group showed a higher protein content (*p* < 0.01) than the control and V75 + B25 groups.

## 4. Discussion

To our knowledge, this is the first study in which butchery wastes have been tested as substrate for Black Soldier fly larvae.

The inclusion of butchery wastes in the different proportions tested in our trial induced high total larval biomass production, expressed on dry matter basis, but also a higher mortality rate of larvae in comparison to the other groups. The best results were obtained when butchery wastes were “diluted” with high proportion of vegetable mix, but vegetable mix, alone, is not suitable for a good larval production. In fact, larvae on V100 diet showed, in general, the worst growing performance (live weight at 22 days, larvae length and thickness, growth rate) even if had a low mortality rate, high values of SRI, PR and a LY not different from the Control and V75 + B25 groups.

A lower growth rate of BSF larvae on V100 diet could be ascribed to both a lower protein availability and a high moisture content. Indeed, considering the protein and moisture percentages of each substrate, the total protein available for larvae growth were: 679.6, 279.8, 543.1, and 482.4 g for control, V100, V50 + B50, and V75 + B25 groups, respectively. Even if the moisture percentage in each substrate were within the suitable range for BSF indicated by Cammack and Tomberlin [28], Dzepe et al. [29], testing five substrates with increasing moisture content from 40 to 80%, observed that increasing the substrate moisture content reduces the larval feed reduction, wet weight, development time, body size, and body thickness. Lalander et al. [30] also reported that high levels of moisture in the substrate reduced the biomass conversion ratio and survival rate of the larvae. However, in our trial, the ventilation applied in the larvae-growing chamber can alleviate the negative effects of high-moisture substrates, according to Pinotti and Ottoboni [31].

Surprisingly, our results showed that the development time of larvae was not different among the groups. This result is in contrast with other researches [7,8,29]. It is not easy to explain this point, thus further insights need to clarify it, evaluating the metabolic profile of larvae in detail.

The V75 + B25 and V100 diet (which larvae showed the second and the first lower amount of total protein) determined a high PR value. These results agree with the findings of Bonelli et al. [32], who showed that the midgut of *H. illucens* larvae can adapt to diets with different nutrient contents, increasing proteolytic activity and decreasing α-amylase and lipase activities when poor diets are available.

The larvae obtained from substrates containing butchery wastes showed a higher percentage of lipids but a lower percentage of proteins than the other groups, and this was particularly true when butchery wastes were used at the highest level. In the larvae, the body fat represents the tissue in which nutrients such as protein, carbohydrates, and fats were stored [33] and used for growth and metamorphosis [34]. However, a high percentage of fat does not indicate a satisfactory accumulation of nutrients reserves [35]. In fact, the lipid tissue of insects is composed of trophocytes, in the cytoplasm of which it is possible to detect two types of roundish structures associated with nutrient accumulation: lipid and protein droplets, differing in terms of size and coloring reactions [35]. The diet containing only 25% of butchery wastes seemed to be more balanced for BSF larvae, as their lipid and protein contents were not different from the control group.

The V100 diet produced larvae with a protein content comparable to the control but with a very low amount of lipids, despite a similar percentage of carbohydrates. This might be surprising, because larvae use free sugars, abundant in the vegetable mix, to produce triacylglycerol, accumulating it in body fat [36]. However, the V100 diet had 25.7% more moisture than the average of the other diets, and this strongly diluted the nutrients available for larval growth.

The average ash content in larvae was higher than in the rearing substrate, and this suggests that larvae accumulate minerals in their body. Indeed, the rate of accumulation was different, according to the used substrate. When a high amount of vegetable mix was included in the diet, as happened in V75 + B25 and V100 groups, the larvae showed 1.38 and 1.30 more ashes than the correspondent diets, respectively. On the contrary, with control and V50 + B50 diets, the rate of ash increase was 1.97 and 2.48, respectively. A possible explanation could be that, in the vegetable mix, some minerals could be complexed with phytates, which reduces the mineral availability for digestion. The effect of phytate on insect growth and development is still poorly investigated [37], but it can reduce the availability of essential minerals and proteins [38]. In addition, phytate in plants plays a defensive role against phytophagous insects, as shown by Green et al. [39], who demonstrated a positive correlation between the presence of phytic acid in the diet and the mortality of three Lepidoptera species. The control diet, consisting of a broiler standard feed, contained wheat and thus an amount of phytate. However, as a commercial diet, it contains a further supplementation of calcium, available phosphorous, and other minerals that may have been easily available for BSF larvae.

Additionally, the protein content in larvae was higher than in the correspondent substrates, ranging from 1.95 of the control diet to 2.51 of the V75 + B25 diet, according to Pinotti and Ottoboni [31]. In our trial, substrates containing the highest CP and moisture percentage (control and V100) allowed to obtain BSF larvae with the highest CP level, according to Meneguz et al. [7].

The evaluation of lipid oxidation is a useful method to measure the integrity of BSF larvae [40]. MDA is one of the most important aldehydes produced during the secondary lipid oxidation of polyunsaturated fatty acids and is considered the major marker for lipid oxidation. Based on the standard values, the BSF larvae of all the tested groups can be considered not rancid (<1.5 mg MDA/kg) [41].

The increased production of CD indicates a major lipid oxidation in the control and V50 + B50 groups [42]. However, the measurement of CD could be interfered with by compounds absorbing in the same region, such as the presence of conjugated double bonds in the original fatty acids [43] or the presence of carotenoids [44].

Concerning the oxidative status of larvae, the lower level of d-ROMs in the hemolymph showed that the use of vegetable waste, at different levels, in the diet of *Hermetia illucens* larvae led to a significant reduction in ROS production. Conversely, the BAP did not show differences among groups, thus suggesting that the vegetable diets did not increase the antioxidant barrier, but some other mechanisms were involved.

In general, oxidative stress can be defined as a disturbance in the balance between the production of reactive oxygen species and antioxidant defenses [45]. In insects, ROS are involved in the regulation of various mechanisms and intercellular signaling and act as bactericidal agents. They can also induce cellular senescence, apoptosis, and cell growth regulatory pathways and are involved in immunity; also, in response to nutrient stress, cells enter autophagy, which can lead to adaptation or death [46]. ROS activation is suspected to serve as a primary mechanism inhibiting development of the pathogen in situ [47]. Since ROS generation in the invertebrate systems may be due to many causes, further studies are needed to explore the mechanisms by which vegetable waste can act as a ROS limiting factor in *Hermetia illucens* diet. However, the higher level of ROS in the control group is not accompanied by a high mortality rate, whereas the latter was higher in the V50 + B50 and V75 + B25 groups. The percentage of survived larvae was, in general, satisfactory, considering that Nguyen et al. [48] found a survival rate of 77% on BSF larvae growing on vegetable wastes. Some authors attributed the low survival rates to the intraspecific competition between individuals for the feed source [49,50] and to the type of substrate [51]. Unfortunately, we cannot be able to evaluate at what stage of larval development the recorded mortalities occurred: this could be very interesting to determine for how long the larvae can be fed with a specific diet.

## 5. Conclusions

The use of butchery wastes as growing substrate for BSF larvae can be suitable, but they must be well combined with other ingredients to balance the high lipid a low protein contents. Vegetable wastes can be appropriate candidates to counteract the negative effects of butchery wastes. The use of vegetable wastes reduces the level of ROS in insect hemolymph, suggesting a positive effect of larvae welfare. However, the diet composed exclusively of vegetable wastes seems to be not indicated for black soldier fly growth as less larval biomass was obtained. Further analyses are in progress at our laboratories to assess the fatty acid, amino acid, and mineral profile of substrates, larvae, and frass.

## Figures and Tables

**Table 1 animals-11-03515-t001:** Substrate chemical composition.

Substrate	Moisture, %	Ash, % DM	Lipids, % DM	Protein, % DM	Carbohydrates, % DM
Control	70.05	3.69	5.12	22.69	68.50
V100	85.37	10.41	3.98	19.11	66.50
V75 + B25	70.13	3.64	36.97	16.15	43.24
V50 + B50	63.65	1.84	49.76	14.88	33.52

V100: total vegetable diet; V75 + B25: diet consisting in 75% of vegetables and 25% of butchery wastes; V50 + B50: diet consisting in 50% of vegetables and 50% of butchery wastes.

**Table 2 animals-11-03515-t002:** Live weight (g) of black soldier fly larvae from 5 to 22 days of age.

	Control	V100	V75 + B25	V50 + B50	RMSE	*p*-Value
5 d	0.0582	0.0586	0.0587	0.0579	0.004	0.8598
7 d	0.0621	0.0627	0.0629	0.0618	0.009	0.9932
9 d	0.1262 ^ab^	0.1092 ^b^	0.1280 ^a^	0.0804 ^c^	0.014	<0.0001
11 d	0.1353 ^b^	0.1413 ^b^	0.1533 ^a^	0.1168 ^c^	0.0095	<0.0001
13 d	0.1365 ^b^	0.1577 ^a^	0.1577 ^a^	0.1353 ^b^	0.013	<0.0001
15 d	0.1549 ^ab^	0.1584 ^a^	0.1636 ^a^	0.1423 ^b^	0.013	0.0045
17 d	0.1653 ^b^	0.1603 ^b^	0.1725 ^ab^	0.1798 ^a^	0.012	0.0033
20 d	0.1913 ^ab^	0.1656 ^b^	0.1971 ^a^	0.1854 ^b^	0.0019	<0.0001
22 d	0.2162 ^a^	0.1839 ^c^	0.2163 ^a^	0.2047 ^b^	0.0078	<0.0001

V100: total vegetable diet; V75 + B25: diet consisting in 75% of vegetables and 25% of butchery wastes; V50 + B50: diet consisting in 50% of vegetables and 50% of butchery wastes; within rows: ^a, b, c^: *p* < 0.01; RMSE: Root Mean Square Error.

**Table 3 animals-11-03515-t003:** Body length (cm) of black soldier fly larvae from 7 to 22 days of age.

	Control	V100	V75 + B25	V50 + B50	RMSE	*p*-Value
7 d	1.14	1.12	1.18	1.16	0.067	0.1199
9 d	1.31 ^b^	1.31 ^b^	1.52 ^a^	1.25 ^b^	0.166	0.0046
11 d	1.31 ^c^	1.39 ^bc^	1.67 ^a^	1.58 ^ab^	0.161	<0.0001
13 d	1.34 ^c^	1.48 ^bc^	1.71 ^a^	1.65 ^ab^	0.151	<0.0001
15 d	1.43 ^c^	1.57 ^bc^	1.77 ^a^	1.67 ^ab^	0.121	<0.0001
17 d	1.44 ^b^	1.63 ^ab^	1.78 ^a^	1.75 ^a^	0.178	0.0003
20 d	1.47 ^b^	1.64 ^ab^	1.84 ^a^	1.77 ^a^	0.210	0.0022
22 d	1.55 ^b^	1.65 ^ab^	1.87 ^a^	1.80 ^a^	0.182	0.0003

V100: total vegetable diet; V75 + B25: diet consisting in 75% of vegetables and25% of butchery wastes; V50 + B50: diet consisting of 50% of vegetables and 50% of butchery wastes; within rows: ^a, b, c^: *p* < 0.01; RMSE: Root Mean Square Error.

**Table 4 animals-11-03515-t004:** Body thickness (cm) of black soldier fly larvae from 7 to 22 days of age.

	Control	V100	V75 + B25	V50 + B50	RMSE	*p*-Value
7 d	0.24 ^b^	0.22 ^b^	0.29 ^a^	0.25 ^ab^	0.034	0.0003
9 d	0.40 ^a^	0.33 ^b^	0.42 ^a^	0.30 ^b^	0.032	<0.0001
11 d	0.42 ^ab^	0.37 ^c^	0.44 ^a^	0.40 ^bc^	0.031	<0.0001
13 d	0.43 ^b^	0.37 ^c^	0.47 ^a^	0.41 ^b^	0.024	<0.0001
15 d	0.43 ^b^	0.37 ^c^	0.47 ^a^	0.42 ^b^	0.025	<0.0001
17 d	0.44 ^b^	0.38 ^c^	0.48 ^a^	0.45 ^ab^	0.032	<0.0001
20 d	0.55 ^a^	0.38 ^c^	0.50 ^b^	0.47 ^b^	0.028	<0.0001
22 d	0.55 ^a^	0.38 ^c^	0.50 ^b^	0.49 ^b^	0.025	<0.0001

V100: total vegetable diet; V75 + B25: diet consisting in 75% of vegetables and 25% of butchery wastes; V50 + B50: diet consisting in 50% of vegetables and 50% of butchery wastes; within rows: ^a, b, c^: *p* < 0.01; RMSE: Root Mean Square Error.

**Table 5 animals-11-03515-t005:** Growth performance of black soldier fly larvae calculated at the end of the trial.

	Control	V100	V75 + B25	V50 + B50	RMSE	*p*-Value
Mortality, %	9.95 ^c^	10.61 ^c^	18.93 ^b^	20.47 ^a^	1.075	<0.0001
TLB, g DM	657.7 ^b^	370.5 ^c^	717.2 ^a^	688.8 ^ab^	18.49	<0.0001
TLF, g DM	1673.2 ^a^	443.2 ^d^	1013.5 ^c^	1464.5 ^b^	87.31	<0.0001
L/H	2.82 ^c^	4.34 ^a^	3.74 ^ab^	3.67 ^b^	0.23	<0.0022
GR	0.011 ^a^	0.009 ^b^	0.011 ^a^	0.010 ^a^	0.0009	<0.0001
SR	44.13 ^c^	69.71 ^a^	66.07 ^b^	59.88 ^b^	3.13	<0.0001
LY	0.22 ^ab^	0.25 ^a^	0.24 ^a^	0.19 ^b^	0.01	<0.0001
WRI	2.60 ^c^	4.10 ^a^	3.89 ^ab^	3.52 ^b^	0.25	<0.0001
ECD	0.48 ^a^	0.36 ^b^	0.36 ^b^	0.32 ^c^	0.015	<0.0001
PR	0.26 ^b^	0.36 ^a^	0.36 ^a^	0.23 ^b^	0.023	<0.0001

V100: total vegetable diet; V75 + B25: diet consisting of 75% of vegetables and 25% of butchery wastes; V50 + B50: diet consisting of 50% of vegetables and 50% of butchery wastes; TLB = total larval biomass; TLF = total larval frass; L/H: length to height ratio; GR = growth rate; SR = substrate reduction; LY = larvae yield; WRI = waste reduction index; ECD = efficiency conversion of digested food; PR: protein conversion ratio. Within rows: ^a, b, c, d^: *p* < 0.01; RMSE: root mean square error.

**Table 6 animals-11-03515-t006:** Hemolymph oxidative stress profile of black soldier fly larvae at 20 days of age.

	Control	V100	V75 + B25	V50 + B50	RMSE	*p*-Value
d-ROMs, U CARR	113.0 ^a^	86.74 ^b^	69.21 ^b^	73.50 ^b^	21.71	<0.0001
BAP, µmol/L	3860.5	3802.1	3967.3	3973.2	767.2	0.9016

V100: total vegetable diet; V75 + B25: diet consisting in 75% of vegetables and 25% of butchery wastes; V50 + B50: diet consisting in 50% of vegetables and 50% of butchery wastes; d-ROMs: Diacron Reactive Oxygen Metabolites; BAP: Biological Antioxidant Potential; within rows: ^a, b^: *p* < 0.01; RMSE: Root Mean Square Error.

**Table 7 animals-11-03515-t007:** Black soldier fly larvae chemical composition and oxidative stability.

	Control	V100	V75 + B25	V50 + B50	RMSE	*p*-Value
Moisture, %	66.22 ^b^	77.46 ^a^	59.10 ^c^	57.69 ^c^	0.29	<0.0001
Ash, % DM	7.28 ^b^	13.50 ^a^	5.05 ^c^	4.57 ^d^	0.55	<0.0001
Lipids, % DM	22.12 ^b^	6.01 ^c^	27.08 ^b^	35.56 ^a^	4.10	<0.0001
Crude protein, % DM	44.24 ^a^	44.87 ^a^	40.67 ^ab^	30.65 ^b^	3.79	<0.0001
CD, mmol Hydroperoxide/100 g	1.64 ^a^	0.56 ^b^	0.86 ^b^	1.69 ^a^	0.30	<0.0001
TBARS, mg MDA-eq/kg	0.18 ^b^	0.22 ^ab^	0.27 ^a^	0.21 ^ab^	0.045	0.0032

V100: total vegetable diet; V75 + B25: diet consisting in 75% of vegetables and 25% of butchery wastes. DM: dry matter; V50 + B50: diet consisting in 50% of vegetables and 50% of butchery wastes; CD: conjugated dienes; TBARS: thiobarbituric acid reactive substances; within rows: ^a, b, c, d^: *p* < 0.01; RMSE: root mean square error.

**Table 8 animals-11-03515-t008:** Frass chemical composition.

	Control	V100	V75 + B25	V50 + B50	RMSE	*p*-Value
Moisture, %	33.18 ^b^	41.63 ^a^	21.28 ^c^	42.39 ^a^	0.31	<0.0001
Ash, % DM	12.57 ^b^	15.13 ^a^	8.67 ^c^	3.11 ^d^	0.89	<0.0001
Lipids, DM	1.77 ^c^	2.47 ^c^	19.98 ^b^	39.32 ^a^	2.41	<0.0001
Protein, % DM	16.60 ^b^	21.72 ^a^	15.65 ^b^	18.98 ^ab^	3.88	<0.0001

V100: total vegetable diet; V75 + B25: diet consisting of 75% of vegetables and 25% of butchery wastes; V50 + B50: diet consisting in 50% of vegetables and 50% of butchery wastes; DM: dry matter; within rows: ^a, b, c, d^: *p* < 0.01; RMSE: root mean square error.

## Data Availability

Data are available to the authors on request.

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
