# Peer review of "Different Combinations of Butchery and Vegetable Wastes on Growth Performance, Chemical-Nutritional Characteristics and Oxidative Status of Black Soldier Fly Growing Larvae"

_animals, 2021, doi:10.3390/ani11123515_

Round 1

Reviewer 1 Report

Animals 2021_1446724

The manuscript entitled with “Exploring new substrates for Black Soldier fly: effect of vegeta-2 ble and butchery wastes on growth performance, chemical-nutritional characteristics and oxidative status of growing larvae” submitted by Nicola Francesco Addeo et al., intends to find better ingredients and suitable combination and the best utilization of dietary substrates to balance the material nutritional requirements of Black Soldier fly. However, this manuscript probably needs a major revision to meet the good standard for publication on Animals.

As we learn from this manuscript, 4 different substrates were used for evaluation regarding their suitability for larva rearing; combination of butchery wastes and or with other ingredients to balance the lipid and protein; vegetable wastes can be an appropriate candidate; vegetable and butchery wastes are easy to find and collect, etc. To be frank, they are not new but to get better mixing fashion or composition optimization. Therefore, the title “Exploring new substrates …” is too fancy to use and should be carefully revised. I recommend the authors to deleted those words. And better to use solid words for a new title to reflect the true contents of the manuscript.

Totally 46 references are found in this submitted manuscript. Only 4 papers from 2021 were cited, and some recent publication has been mentioned actually, e.g. Lu et al., Effects of Different Nitrogen Sources and Ratios to Carbon on Larval Development and Bioconversion Efficiency in Food Waste Treatment by Black Soldier Fly Larvae (Hermetia illucens). Insects, 2021, 12:507. Good to include the related studies published in this year as well.

As for the Results, obviously, the overall writing style and sentences, forgive me to say so, are so simple, every paragraph starts with “Table X reports ……”, truly lacking of variability and impressiveness.

As for the Discussion, obviously, it is too long! Energy demanding and time consuming, for author to write and reader to read! I would say to let long paragraph short, since it could be just too good to conclude two or several key points to be well focused. The major results should be discussed in an adequate way. From Line 362 to Line 371, why the authors preferred using only single sentence as a separate paragraph? Is the meaning completed or just too important?

Some minor issues:

Line 16, after the “to be very promising ”, missing a period. It should be added as “to be very promising.”

Line 17, “In this context”. It is better to put a comma after the word “context”.

Lines 137 – 149, in section of 2.2 Growing Trial, Please try to use equation or formula to describe these contents. More scientific and concise!

Line 505, “Retrieved September 8, 2021, from ….”   What is this?

Some tiny typos need to revise.

Author Response

We are grateful for your and the reviewer’s comments, and the evaluation of our work. We have revised and modified the text according to the referees' critiques. Therefore, we have provided new information about the general aim of the trial and added many new and clarifying statements in all parts of the paper. We hope that these changes have improved the manuscript considerably and we hope that it can be published.

Reviewing the manuscript, we found an error in calculation of ECD reported in Table 5. Now the data were corrected.

Reviewer 1

As we learn from this manuscript, 4 different substrates were used for evaluation regarding their suitability for larva rearing; combination of butchery wastes and or with other ingredients to balance the lipid and protein; vegetable wastes can be an appropriate candidate; vegetable and butchery wastes are easy to find and collect, etc. To be frank, they are not new but to get better mixing fashion or composition optimization. Therefore, the title “Exploring new substrates …” is too fancy to use and should be carefully revised. I recommend the authors to deleted those words. And better to use solid words for a new title to reflect the true contents of the manuscript.

Au: thank you very much for your comment. The title has been modified.

Totally 46 references are found in this submitted manuscript. Only 4 papers from 2021 were cited, and some recent publication has been mentioned actually, e.g. Lu et al., Effects of Different Nitrogen Sources and Ratios to Carbon on Larval Development and Bioconversion Efficiency in Food Waste Treatment by Black Soldier Fly Larvae (Hermetia illucens). Insects2021, 12:507. Good to include the related studies published in this year as well.

Au: the suggested reference has been added and other more recent articles have been cited

 As for the Results, obviously, the overall writing style and sentences, forgive me to say so, are so simple, every paragraph starts with “Table X reports ……”, truly lacking of variability and impressiveness.

Au: you are right! Sometimes simplicity is better, sometimes no! The beginning of the paragraphs has been modified

 As for the Discussion, obviously, it is too long! Energy demanding and time consuming, for author to write and reader to read! I would say to let long paragraph short, since it could be just too good to conclude two or several key points to be well focused. The major results should be discussed in an adequate way.

Au: The discussion has been shortened and repetitions have been deleted

From Line 362 to Line 371, why the authors preferred using only single sentence as a separate paragraph? Is the meaning completed or just too important?

 Au: now they are a single paragraph.

 Some minor issues: Line 16, after the “to be very promising ”, missing a period. It should be added as “to be very promising.”

Au: a dot has been added

Line 17, “In this context”. It is better to put a comma after the word “context”.

Au: done

Lines 137 – 149, in section of 2.2 Growing Trial, Please try to use equation or formula to describe these contents. More scientific and concise!

Au: formulae have been rewritten

Line 505, “Retrieved September 8, 2021, from ….”   What is this?

Au: corrected

Some tiny typos need to revise.

Au: corrected

Reviewer 2 Report

1 One of the main questions confuses me is that what is the goal of this work. To degrade butchery waste? to obtain high quality larvae? to obtain larvae lipid? to obtain larvae protein? Which one? If the aim is just to test the best vegetable to butchery waste ratio, it is too narrow, isn’t it? Since the goal is missing, it is hard to tell you succeed or not.    

2. What is the universal message that could be useful to the people who work in this field? In another word, if you change the substrate, change the vegetable to butchery waste ratio of the substrate, everything will be changed. If so, what is the common results can be speculated from your work. This should be explored in the discussion part.       

3. The title is not accurate, it is not exploring new substrate, but exploring new component ratio. 

4. Why use butchery for BSF substrate? There is no reason in the introduction part. And butchery waste could be highly diverse all around the word. Butchery waste could contain high amount of animal tissues or organs, but this kind of butchery waste mainly contains bovine fat. BSFL is good for the degradation of protein or carbohydrate but not for fat. The author needs to state why use fat for BSFL treatment, isn’t it better to refine the fat prior to feeding BSFL.     

5. Line99, if 5-DOL is the abbreviation of “five days old black soldier fly larvae”, it is crazy. Please think about changing it.   

6. Line 117, how to weight 6000 larvae, please add details.   

7. Line 137-149, the formula should be input as formulas but not text. This way of presenting formula is crazy. Calculation is based on wer wt or dry wt? The last formular using PCR as the abbreviation of protein conversion ratio is not appropriate. PCR is actually short of polymerase chain reaction.     

8. Line154, 158, it is not appropriate to cite methods only with reference numbers without brief description. It is hard to guess what is exactly done.    

9. Line 187, this formula is not necessary, hard to tell where it is right.   

10. For all the tables, the samples were listed as the order of control, V100, V50+B50, V75+B25, it is quite disturbing for me, that I expected the order to be control, V100, V75+B25, V50+B50, similar to the sequence of growing butchery waste percentage. I hope the author could change the order of samples to make it easier for the readers.  

11. Line208, the statistics was based on rows or columns?   

12. Table5, why there is no calculation of larvae yields, i.e. how much larvae dry wt produced from waste dry wt? Again the goal of this work is missing. To have low mortality, high GR, SR, WRI, ECD or PCR? Without larvae yields, it is hard to tell which condition is the best.  

13. Table7, what is your goal? High lipid content? High protein content? High lipid or protein conversion ratio?      

14. From the Table 1 and Table 7, it is easy to tell that the more butchery you added, the higher lipid and lower protein entered into your substrate, as BSFL like protein and dislike lipid, the good results will be shown in the substrate contains higher protein. This mechanism is simple and clear. However, the discussion part tried to make this simple mechanism complicated. Table 2-8, they are all the results, only Table 1 is the reason. Even Table 6 and 7, they are still results. Why not dig the relationship between the reasons and results, instead of using results (Table6,7) to explain results. More statistical analysis could be done between table 1 and table 2-8, such as PCA, correlations or regressions. 

Author Response

We are grateful for your and the reviewer’s comments, and the evaluation of our work. We have revised and modified the text according to the referees' critiques. Therefore, we have provided new information about the general aim of the trial and added many new and clarifying statements in all parts of the paper. We hope that these changes have improved the manuscript considerably and we hope that it can be published.

Reviewing the manuscript, we found an error in calculation of ECD reported in Table 5. Now the data were corrected.

Reviewer 2

1 One of the main questions confuses me is that what is the goal of this work. To degrade butchery waste? to obtain high quality larvae? to obtain larvae lipid? to obtain larvae protein? Which one? If the aim is just to test the best vegetable to butchery waste ratio, it is too narrow, isn’t it? Since the goal is missing, it is hard to tell you succeed or not.

Au: you're right, the goal of the research was written a bit 'vague. We have reformulated it; we hope that now the text is clearer than before.

  1. What is the universal message that could be useful to the people who work in this field? In another word, if you change the substrate, change the vegetable to butchery waste ratio of the substrate, everything will be changed. If so, what is the common results can be speculated from your work. This should be explored in the discussion part.

Au: the discussion has been modified in particular in the first part, focusing on the purpose of the research.

  1. The title is not accurate, it is not exploring new substrate, but exploring new component ratio.

Au: thank you very much, the title has been changed

  1. Why use butchery for BSF substrate? There is no reason in the introduction part. And butchery waste could be highly diverse all around the word. Butchery waste could contain high amount of animal tissues or organs, but this kind of butchery waste mainly contains bovine fat. BSFL is good for the degradation of protein or carbohydrate but not for fat. The author needs to state why use fat for BSFL treatment, isn’t it better to refine the fat prior to feeding BSFL.

Au: you are right, and also vegetable wastes could be very different among Countries, season, etc. This is, we suppose, for all kinds of wastes. However, we have added some lines in the introduction to justify the use of these kinds of substrates.

  1. Line99, if 5-DOL is the abbreviation of “five days old black soldier fly larvae”, it is crazy. Please think about changing it.

Au: tha achronymus has been deleted

  1. Line 117, how to weight 6000 larvae, please add details.   

Au: more details were added

  1. Line 137-149, the formula should be input as formulas but not text. This way of presenting formula is crazy. Calculation is based on wer wt or dry wt? The last formular using PCR as the abbreviation of protein conversion ratio is not appropriate. PCR is actually short of polymerase chain reaction.

Au: formulae have been rewritten. The weights are on dry matter basis (indicated in the text). The acronym PCR has been changed as PR

  1. Line154, 158, it is not appropriate to cite methods only with reference numbers without brief description. It is hard to guess what is exactly done.    

Au: more details have been added for the used methods

  1. Line 187, this formula is not necessary, hard to tell where it is right.

Au: The formula has been deleted

  1. For all the tables, the samples were listed as the order of control, V100, V50+B50, V75+B25, it is quite disturbing for me, that I expected the order to be control, V100, V75+B25, V50+B50, similar to the sequence of growing butchery waste percentage. I hope the author could change the order of samples to make it easier for the readers.

Au: the tables and text (M&M) have been modified according to your suggestion

  1. Line208, the statistics was based on rows or columns?

Au: added

  1. Table5, why there is no calculation of larvae yields, i.e. how much larvae dry wt produced from waste dry wt? Again the goal of this work is missing. To have low mortality, high GR, SR, WRI, ECD or PCR? Without larvae yields, it is hard to tell which condition is the best.

Au: thank you for your comment, larvae yield has been calculated and added

  1. Table7, what is your goal? High lipid content? High protein content? High lipid or protein conversion ratio?

Au: the aim of the research has been more clearly indicated at the end of the introduction

  1. From the Table 1 and Table 7, it is easy to tell that the more butchery you added, the higher lipid and lower protein entered into your substrate, as BSFL like protein and dislike lipid, the good results will be shown in the substrate contains higher protein. This mechanism is simple and clear. However, the discussion part tried to make this simple mechanism complicated. Table 2-8, they are all the results, only Table 1 is the reason. Even Table 6 and 7, they are still results. Why not dig the relationship between the reasons and results, instead of using results (Table6,7) to explain results. More statistical analysis could be done between table 1 and table 2-8, such as PCA, correlations or regressions. 

Au: thank you for your comment. We’re working on another manuscript in which the correlations will be displayed. Surely our results are determined by the characteristics of the substrates, this happens in all the trials in which different substrates were used, but, before to study the correlations, in our opinion is important to show the “crude” results as butchery wastes are not yet tested in larvae nutrition, standing our knowledge.

Reviewer 3 Report

The research is interesting and conducted well by the authors. Moreover, the use of butchery waste as a new feeding substrate is an added value to the study since it is the first research on this waste. Results are structured well but the adding of some parameters could complete the research and give further information about the bioconversion process and its products. In the whole text there are minor English error spells, please check the manuscript carefully.

SIMPLE SUMMARY

Line 15-16: “Due to the high sustainability of the insect farming, the possibility to farm insects  as food and feed source seems to be very promising” Please, add a dot next “promising”

Line 19: “…were evaluated for their suitability for larva rearing” Please, replace “larva” with “larvae”

INTRODUCTION

Line 68 the first time you mention H. illucens, I suggest to briefly state the importance of this species, that it the most reared in the world, not only for the bioconversion ability but also for the biological active molecules that can derive from it (for example chitin - doi: 10.3390/cosmetics8020040; doi:10.1002/jctb.6533,  lipids - doi:10.3390/su131810198, AMPs - 10.1007/s00018-021-03784-z, 10.1038/s41598-020-74017-9).

MATERIAL AND METHODS

Line 132-133: “Feeding of larvae was continued until more than 25% of the larvae in a tray had developed into prepupae” How did the authors measure the 25% of prepupae?

RESULTS

Line 193 How did the authors explain the same duration of larval growth (stopped when the 25% of the larvae reached the prepupal stage) between control diet and other substrates? Literature data reported a brief developmental time in larvae fed on control diets.

Line 202: What do the authors mean with “live weight of BSF larvae”?  Is it the live weight of a single larva or is it about the group of larvae measured every two days? It would be better to specify.

Line 268: Larval frass is recently used as organic fertilizer with important results. Could you provide an analysis of the N-P-K ratio? It would be interesting to give a further value to this BSF product.

DISCUSSION                      

Line 313: Since “total larvae biomass” is a trial result, can the authors make a table of this parameter in the “Results” paragraph, as they did with other results? Moreover, did the authors also measure total larval frass biomass? Please, could the authors add this parameter to their results?

Line 360: “The diet containing only 25 of butchery wastes…”

Please, add “%” next to “25”

Author Response

We are grateful for your and the reviewer’s comments, and the evaluation of our work. We have revised and modified the text according to the referees' critiques. Therefore, we have provided new information about the general aim of the trial and added many new and clarifying statements in all parts of the paper. We hope that these changes have improved the manuscript considerably and we hope that it can be published.

Reviewing the manuscript, we found an error in calculation of ECD reported in Table 5. Now the data were corrected.

Reviewer 3

The research is interesting and conducted well by the authors. Moreover, the use of butchery waste as a new feeding substrate is an added value to the study since it is the first research on this waste. Results are structured well but the adding of some parameters could complete the research and give further information about the bioconversion process and its products. In the whole text there are minor English error spells, please check the manuscript carefully.

Au: thank you for your comments

SIMPLE SUMMARY

Line 15-16: “Due to the high sustainability of the insect farming, the possibility to farm insects  as food and feed source seems to be very promising” Please, add a dot next “promising”

Au: done

Line 19: “…were evaluated for their suitability for larva rearing” Please, replace “larva” with “larvae”

Au: done

INTRODUCTION

Line 68 the first time you mention H. illucens, I suggest to briefly state the importance of this species, that it the most reared in the world, not only for the bioconversion ability but also for the biological active molecules that can derive from it (for example chitin - doi: 10.3390/cosmetics8020040; doi:10.1002/jctb.6533,  lipids - doi:10.3390/su131810198, AMPs - 10.1007/s00018-021-03784-z, 10.1038/s41598-020-74017-9).

Au: thank you very much for your comment we added some rows on this.

MATERIAL AND METHODS

Line 132-133: “Feeding of larvae was continued until more than 25% of the larvae in a tray had developed into prepupae” How did the authors measure the 25% of prepupae?

Au: the method has been indicated.

RESULTS

Line 193 How did the authors explain the same duration of larval growth (stopped when the 25% of the larvae reached the prepupal stage) between control diet and other substrates? Literature data reported a brief developmental time in larvae fed on control diets.

Au: some lines have been added on this point at the beginning of the Discussion section

Line 202: What do the authors mean with “live weight of BSF larvae”?  Is it the live weight of a single larva or is it about the group of larvae measured every two days? It would be better to specify.

Au: this point has been clarified

Line 268: Larval frass is recently used as organic fertilizer with important results. Could you provide an analysis of the N-P-K ratio? It would be interesting to give a further value to this BSF product.

Au: thank you for your comment. Unfortunately, it is not possible to provide this analysis for this trial. We surely will evaluate this aspect in a further trial.

DISCUSSION                      

Line 313: Since “total larvae biomass” is a trial result, can the authors make a table of this parameter in the “Results” paragraph, as they did with other results? Moreover, did the authors also measure total larval frass biomass? Please, could the authors add this parameter to their results?

Au: missing data have been added

Line 360: “The diet containing only 25 of butchery wastes…”

Please, add “%” next to “25”

Au: done

Round 2

Reviewer 1 Report

Thank you for your responses. Good to have a chance to re-review your manuscript.

The comments have been responded well. Quite many issues have been revised properly.

One last issue is that those super long paragraphs in the Introduction and the Discussion Sections, cause much confusing or misunderstanding to me and readers.

They are really long but contain several layers of meaning. Why not separate into 2-3 well-focused paragraphs?

This probably one priority of the last few things to do. No more further request.

Author Response

Thank you for your responses. Good to have a chance to re-review your manuscript.

Au: thank you to you for your comments that helped improve our manuscript

The comments have been responded well. Quite many issues have been revised properly.

Au: thank you, we are happy with this

One last issue is that those super long paragraphs in the Introduction and the Discussion Sections, cause much confusing or misunderstanding to me and readers.

They are really long but contain several layers of meaning. Why not separate into 2-3 well-focused paragraphs?

This probably one priority of the last few things to do. No more further request.

Au: we divided the introduction and the discussion into smaller paragraphs and we tried to focus each paragraph on a specific topic

Reviewer 2 Report

No more comments.

Author Response

No more comments.

Au: thank you